# Changes in Apathy, Depression, and Anxiety in Parkinson’s Disease from before to during the COVID-19 Era

**DOI:** 10.3390/brainsci13020199

**Published:** 2023-01-24

**Authors:** Shraddha B. Kinger, Truley Juneau, Rini I. Kaplan, Celina F. Pluim, Joshua T. Fox-Fuller, Timothy Wang, Nishaat Mukadam, Sandy Neargarder, Robert D. Salazar, Alice Cronin-Golomb

**Affiliations:** 1Department of Psychological and Brain Sciences, Boston University, Boston, MA 02215, USA; 2Department of Psychology, Bridgewater State University, Bridgewater, MA 02325, USA

**Keywords:** apathy, depression, anxiety, Parkinson’s Disease, COVID-19

## Abstract

Apathy, depression, and anxiety are common non-motor symptoms of Parkinson’s disease (PD). Tracking the changes in such symptoms over time would be valuable not only to determine their natural course during the disease, but also to establish the effects of unusual historical events interacting with the natural course. Having collected data on apathy (Apathy Scale), depression (Beck Depression Inventory-II), and anxiety (Parkinson’s Anxiety Scale) in a large sample of persons with PD (PwPD) before the beginning of the COVID-19 era, we followed up with these individuals to investigate the changes in their prevalence of apathy, depression, and anxiety across two timepoints (T1 and T2). Of the original 347 participants, 111 responded and provided complete data at T2. The data collection at T1, before COVID-19, occurred between 2017–2018. The data collection at T2 occurred in 2021 and included the same measures, with the addition of the Coronavirus Impact Scale to assess the effects of the pandemic on the individual participants. Over this period, there was a significant increase in apathy, but not in depression or anxiety. Anxiety and depression, but not apathy, were correlated with the impact of COVID-19.

## 1. Introduction

Apathy is a non-motor symptom that is often observed in persons with Parkinson’s Disease (PwPD). Its features include a lack of motivation, reduced emotional expression, loss of interest, decreased goal-directed behavior, and indifference that are not attributable to emotional distress or cognitive impairment [1,2,3,4,5]. The prevalence and significance of apathy in PD has been under debate, as the upper range significantly differs across studies. In their systematic review, in 2015, den Brok and colleagues [1] cited 23 papers with a prevalence range between 16.5% and 60%, with the meta-analysis showing a prevalence of 39.8%. Pluck and Brown [4] put this estimate between 16.5% and 42% based on their review of the existing literature at the time (2002). 

Depression includes many of the same symptoms as apathy, making it difficult for researchers to determine whether apathy in an individual is a stand-alone disorder, an aspect of depression [2], or secondary to depression [1]. Studies have demonstrated that apathy can occur independently of depression, and depression independently of apathy, but also that the symptoms may be related in individuals [2,4]. For example, Ou and colleagues [5] found that depression in the early stages of PD is a predictor of later apathy. By contrast, Meyer and colleagues [3] did not find an association between apathy and depression in PwPD. Similarly, Pluck and Brown [4] found no association between apathy and depression scores. This discrepancy in findings may be due, in part, to differences in the measures used, as Meyer et al. [3] and Pluck and Brown [4] used the Beck Depression Inventory-II to assess depression symptoms in their participants, whereas Ou et al. [5] used the Hamilton Depression Rating Scale. From a clinical standpoint, it is important to distinguish apathy from depression in PwPD because apathy may be misdiagnosed as depression [1], and antidepressant medications do not always work for apathy; indeed, at times, antidepressant medications may even worsen apathy symptoms [6]. However, apathy may improve with cognitive-behavioral therapy for depression in PD [7,8].

Anxiety is another common non-motor symptom of PD, with estimates of 31–49% of PwPD experiencing anxiety [9,10,11]. In contrast to depression, anxiety is not clearly associated with apathy [4]. Pluck and Brown [4] observed no differences in the levels of anxiety between high apathy and low apathy groups in their study, and Bogdanova and Cronin-Golomb [9] reported that apathy and anxiety correlated with scores on different cognitive tests in PwPD with left- vs. right-side of motor onset, although both were associated with disease duration. Although there is more evidence for a relation between apathy and depression than a relation between apathy and anxiety, depression and anxiety often co-occur in PD [7,8].

Apathy has been associated with older age, cognitive impairment, longer disease duration, and more severe disease in PwPD [1,5]. In a longitudinal study, Ou and colleagues [5] found that apathy increased over time (4 years). Regarding gender differences in apathy prevalence, the PD literature has been inconclusive, with some results showing no difference [4], and others showing a different pattern for men and women; for example, apathy was correlated with age positively in men but negatively in women [3].

There is currently no consistent picture of apathy in PD, its natural course over the disease and what it is related to, including other prevalent non-motor symptoms. With the COVID-19 pandemic, we have been able to consider these issues as related to a single worldwide historical event that undoubtedly affected all PwPD, albeit to different degrees. Outcomes related to PD have been shown to span a range regarding hospitalization and mortality, although it is not clear whether PD itself increased the prevalence of negative outcomes, rather than associated variables such as advanced age or comorbidities, and it should be noted that most of the available data were from the pre-vaccination era [12]. A recent systematic review of psychiatric disorders in PwPD associated with the COVID-19 era, including apathy, depression, and anxiety, indicated variable findings across mostly cross-sectional studies [13]. In the present two-timepoint longitudinal study of PwPD, we assessed apathy, as well as depression and anxiety, before the pandemic, and again, more recently, during the pandemic.

## 2. Materials and Methods

The Boston University Online Survey Study of Parkinson’s Disease (BOSS-PD) was conducted to assess the experiences of 347 PwPD between June 2017 and December 2018 (timepoint 1, T1). The participants were recruited through advertisements in Fox Trial Finder, Clinicaltrials.gov, American Parkinson Disease Association websites, and other community outreach. The study and participants are described in Islam et al. [14]. The inclusion criteria for PwPD were the diagnosis of PD without dementia, 40+ years of age, 8+ years of education, proficient in English, with access to a computer. The exclusion criteria were active neoplasm, serious cardiac disease, other serious chronic medical illness, prior intracranial surgery, history of traumatic brain injury, psychiatric or neurological diagnoses other than PD, history of alcoholism or other drug abuse, or treatment with electroconvulsive therapy. 

For the follow-up survey, PD-COVID, we attempted to recontact all of the BOSS-PD participants by email in order to assess how this cohort was faring during the pandemic. We received 130 responses between March and June 2021. The data from the 19 individuals who did not complete all of the relevant questionnaires were excluded, for a final sample of 111 PwPD (timepoint 2, T2). The survey included most of the same measures as the BOSS-PD, plus new measures, including the Coronavirus Impact Scale (CIS).

Both the BOSS-PD and PD-COVID were online Qualtrics-based surveys. Informed consent was obtained from all of the participants involved in the study before each survey was made available. All of the procedures were approved by the Institutional Review Board of Boston University. The measures used in the present study are listed below. Each was administered at both timepoints, except for the Coronavirus Impact Scale, which was given only at T2. Additionally, we administered the Movement Disorders Society-Unified Parkinson’s Disease Rating Scale [15] Parts I and II, measuring non-motor (UPDRS_non-motor_) and motor (UPDRS_motor_) experiences of daily living, respectively, as an index of subjective disease severity. Higher scores indicate more severe symptoms.

### 2.1. Coronavirus Impact Scale (CIS)

The Coronavirus Impact Scale is a 12-item self-report measure that assesses multiple aspects of living during the COVID-19 pandemic. Each of the first nine items is considered on a four-point scale (none, mild, moderate, severe). The first eight items assess a variety of COVID-19 impacts, including changes in routines; family income/employment; food access; medical health care access; mental health treatment access; access to extended family and non-family social supports; experiences of stress related to the pandemic; and stress/discord in the family. Item #9 is about personal diagnosis of coronavirus; #10 about diagnosis in immediate family; #11 about diagnosis in extended family and close friends. Items #10 and #11 have a five-point scale with the last being deaths due to coronavirus. The twelfth item is an open-ended response asking participants for other ways in which COVID-19 has impacted their lives [16]. Higher scores indicate greater COVID-19 impact.

### 2.2. Apathy Scale (AS)

The AS is a 14-item self-report measure that assesses apathy on a scale of 0–3. For the first 8 questions, a score of 3 means “not at all”, 2 means “slightly”, 1 means “some”, and 0 means “a lot”. For questions 9–14, these scores are reversed. Higher scores indicate higher levels of apathy [17].

### 2.3. Beck Depression Inventory-II (BDI-II)

The BDI-II is a 21-item self-report measure that assesses the severity of depressive symptoms on a scale of 0–3, with higher scores indicating more severe symptoms [18,19]. 

### 2.4. Parkinson’s Anxiety Scale (PAS)

The PAS is a 12-item measure that assesses anxiety symptoms. Each item is assessed on a 5-point scale with “0” meaning “not or never” and “5” meaning “severe or almost always” [20]. 

### 2.5. Data Analysis

Two-tailed paired-samples *t*-tests were used to compare the demographic, clinical, and psychiatric variables of the sample at T1 and T2. We conducted Pearson correlations between the scores on the psychiatric measures (apathy, depression, anxiety) and other variables of interest (at T1 and T2), including age, education, disease duration, disease severity (non-motor and motor), and time elapsed between timepoints for individual participants. We also conducted Pearson correlations between the scores on the Coronavirus Impact Scale and the same variables of interest at T2. The Pearson correlations were also conducted among the psychiatric variables at T1 and T2. As the prevalence of the psychiatric disorders may differ for men and women with PD, we compared the men and women on the variables of interest and examined the gender effects by conducting all of the same correlations for women and men separately. Spearman correlations were used to examine the relation between the apathy scores (the primary variable of interest) and each of the nine scaled CIS items for the entire group and for women and men separately, with Bonferroni correction to account for the multiple comparisons.

## 3. Results

### 3.1. Participants

A total of 111 participants (50 men, 61 women) responded to all of the relevant questionnaires, including the Apathy Scale in both the BOSS-PD and PD-COVID surveys. Table 1 provides the means and standard deviations (SD) of the demographic, clinical, and performance variables for the entire group and for men and women separately. All of the participants identified as non-Hispanic and White.

At T1, the ages ranged between 40 and 81 years and at T2, between 44 and 84 years. The education level was the same at both timepoints, ranging between 10 and 21 years. At T1, disease duration ranged between 0 and 37 years and at T2, between 2.6 and 40.3 years. At T1, the UPDRS_non-motor_ scores ranged between 0 and 18 and at T2, between 0 and 21. At T1, the UPDRS_motor_ scores ranged between 0 and 27 and at T2, between 0 and 33. The CIS was administered at T2 only. The scores ranged between 9 and 26, with a mean of 16.4 (SD 3.5), indicating mild to moderate impact of the pandemic. 

There were no significant differences between men and women for age, education, UPDRS_non-motor_, UPDRS_motor_, apathy, depression, or anxiety at T1 or T2 (all t < 1.61, all *p* > 0.11). For disease duration, there was a trend for women to have longer duration than men (T1 t(109) = 2.0, *p* = 0.52; T2 t(109) = 1.9, *p* = 0.59). Women had a significantly higher total CIS score than men (means of 17.0 and 15.7, respectively) (t(109) = 2.0, *p* = 0.048).

Table 2 provides information on those participants in the BOSS-PD who did not complete the PD-COVID (referred to as “Only BOSS-PD”), for comparison purposes.

### 3.2. Apathy

The paired-samples t-tests demonstrated a significant increase in apathy from T1 (M = 8.8, SD = 6.4) to T2 (M = 11.2, SD = 6.1) (t(110) = 4.4, *p* < 0.001, *d* = 0.42). This increase occurred for both men (T1 M = 9.6, SD = 5.8; T2 M = 12.2, SD = 6.7 (t (49) = 3.1, *p* = 0.003, *d* = 0.44) and women (T1 M = 8.2, SD = 6.9; T2 M = 10.3, SD = 5.5, t (60) = 3.1, *p* = 0.003, *d* = 0.40). The effect size (*d*) in each case was close to the conventional value of 0.5 that is interpreted as moderate [21].

In the whole group, at neither T1 nor T2 was there a significant correlation between the apathy score and age, education, disease duration (all r < 0.18, all *p* > 0.05), nor, at T2, score on the CIS (r = 0.15, *p* = 0.11). These correlations did not differ when analyzing the data separately for men (all r < 0.19, all *p* > 0.05) or women (all r < 0.20, all *p* > 0.05). At T1, the apathy score correlated with PD severity, as indexed by UPDRS_non-motor_ (r = 0.33, *p* < 0.001) and UPDRS_motor_ (r = 0.22, *p* = 0.022); at T2, the correlation of apathy with UPDRS_non-motor_ was only at a trend level (r = 0.17, *p* = 0.076) and the correlation of apathy with UPDRS_motor_ did not hold at all (r = 0.13, *p* = 0.161). For men, at T1, there was a trend towards a significant correlation between apathy and UPDRS_non-motor_ (r = 0.24, *p* = 0.09), but no correlation between apathy and UPDRS_motor_ (r = 0.14, *p* = 0.34). For women, at T1, the apathy score correlated with PD severity as indexed by UPDRS_non-motor_ (r = 0.40, *p* < 0.001) and UPDRS_motor_ (r = 0.26, *p* = 0.04). At T2, for men, the apathy score was not correlated with PD severity for either UPDRS_non-motor_ or UPDRS_motor_ (all r < 0.12, all *p* > 0.05). For women, at T2, the apathy score showed a trend towards a correlation with UPDRS_non-motor_ (r = 0.23, *p* = 0.08), with no correlation between apathy and UPDRS_motor_ (r = 0.15, *p* = 0.30). Regarding the correlations with the pandemic impact at T2, there was no significant correlation between the apathy scores and total CIS scores in women (r = 0.10, *p* = 0.46), although there was a trend for men (r = 0.27, *p* = 0.059). We also conducted Spearman correlations between the apathy scores and each of the nine scaled CIS items for the entire group and for women and men separately. No correlation was significant after applying Bonferroni correction for multiple comparisons (alpha of 0.05/9 = 0.0056; all r < 0.327, all *p*
> 0.010).

### 3.3. Depression

The paired-samples *t*-tests demonstrated no significant differences between the BDI-II scores at T1 (M = 7.4, SD = 6.5) and T2 (mean = 8.0, SD = 7.0) (t(110) = 1.15, *p* = 0.25) in the whole group. No significant differences were observed for men from T1 (M = 6.8, SD = 4.6) to T2 (M = 8.3, SD = 7.2) (t(49) = 1.5, *p* = 0.12). Similarly, no significant differences were observed for women from T1 (M = 7.8, SD = 7.8) to T2 (M = 7.7, SD = 6.8 (t(60) = 0.11, *p* = 0.92). At neither T1 nor T2 was there a significant correlation between the depression scores and age or disease duration for the whole group (all r < 0.11, all *p* > 0.05), nor for men (all r < 0.12, all *p* > 0.05) or women (all r < 0.13, all *p* < 0.05). A significant negative correlation was found between the education and BDI-II scores at T1 (r = −0.26, *p* = 0.006), but not at T2 (r = −0.12, *p* = 0.22). For men, at neither T1 or T2 was there a significant correlation between the education and BDI-II scores (all r < 0.28, all *p* > 0.05). A significant negative correlation was noted between the education and BDI-II scores for women at T1 (r = −0.26, *p* = 0.04), but not at T2 (r = −0.15, *p* = 0.26). There was a positive correlation between depression and the scores on the CIS at T2 in the whole group (r = 0.35, *p* < 0.001), meaning that more depression was associated with higher COVID impact. This was also found separately for men (r = 0.29, *p* = 0.04) and women (r = 0.44, *p* < 0.001).

### 3.4. Anxiety

The paired-samples t-tests demonstrated no significant differences between the PAS scores at T1 (M = 5.6, SD = 6.0) and T2 (M = 5.5, SD = 6.4) (t(110) = 0.08, *p* = 0.94). No significant differences were observed for men from T1 (M = 4.9, SD = 5.0) to T2 (M = 5.1, SD = 6.5) (t(49) = 0.31, *p* = 0.76) or for women from T1 (M = 6.1, SD = 6.6) to T2 (M = 5.9, SD = 6.4) (t(60) = 0.34, *p* = 0.73). At neither T1 nor T2 was there a significant correlation between the anxiety score and age (all r < 0.17, all *p* > 0.05). A significant negative correlation was found between the education and PAS scores at T1 (r = −0.23, *p* = 0.02), but not at T2 (r = −0.12, *p* = 0.20). For men, at neither T1 or T2 was there a significant correlation between the education and PAS scores (all r < −0.15, all *p* > 0.05). A significant, negative correlation was noted between the education and PAS scores for women at T1 (r = −0.30, *p* = 0.02), but not at T2 (r = −0.10, *p* = 0.45). There was a positive correlation between the PAS scores and disease duration at T1 (r = 0.24, *p* = 0.011) and at T2 (r = 0.24, *p* = 0.012) for the whole group, which was driven by women (all r < 0.35, all *p* < 0.006) but not men (all r < 0.10, all *p* > 0.05). There was also a positive correlation between the scores on the PAS and the CIS at T2 in the whole group (r = 0.28, *p* = 0.003), again driven by women (r = 0.34, *p* = 0.009) and not men (r = 0.21, *p* = 0.15).

### 3.5. Correlations: Apathy, Depression, and Anxiety

There was a significant positive correlation between apathy (AS) and depression (BDI-II) at both T1 (r = 0.48) and T2 (r = 0.56) for the whole group, and these correlations were also significant for women and for men separately. There was also a significant positive correlation between apathy and anxiety (PAS) at both T1 (r = 0.43) and T2 (r = 0.43) for the whole group, as well as for women and men separately. A significant positive correlation between depression and anxiety was similarly found at both T1 (r = 0.64) and T2 (r = 0.71) for the whole group, and for women and men separately. The details are provided in Table 3.

To calculate the amount of time between T1 and T2, the date of completion for T1 was subtracted from the date of completion for T2, and this value was reported in years (M = 3.5, SD = 0.4). There was no significant correlation between this amount of time and the scores at T2 for apathy, depression, or anxiety in the whole group, in men, or in women (all r ≤ 0.216, all *p* ≥ 0.132).

## 4. Discussion

Over the course of a four-year period, with the first survey conducted before the COVID-19 pandemic and the second conducted during it, participants with PD endorsed a significant increase in apathy but not in depression or anxiety. The scores on the measures of all three psychiatric symptoms were correlated at both timepoints, possibly indicating general disease severity or the common comorbidity of these symptoms in general. The increase in apathy over time in PwPD, specifically across a period of about four years, is consistent with that reported by Ou and colleagues in the pre-pandemic period [5]. These investigators reported that apathy increased in prevalence, but not consistently, in individuals across their four timepoints. Similarly to our study, they found an association of apathy with depression (whereas anxiety was not examined). Similarly, Pluim and colleagues recently reported an increase in apathy in PwPD over a two year time period and found an association of apathy with depression [22]. We further found that a greater impact of the pandemic, as assessed with the Coronavirus Impact Scale, was associated with more depression (in men and women) and anxiety (in women), but not with more apathy, although there was a trend in that direction for men.

There are a number of potential reasons for the dissociation in the direction of severity over time for apathy vs. depression and anxiety. First, for many months of the pandemic, people were required to stay at home and isolate themselves from others. Isolation was particularly emphasized for people with additional underlying health conditions and older individuals, which would include most PwPD. D’Iorio and colleagues [23] examined a sample of 42 participants with PD who did not experience an increase in apathy or anhedonia across the pre/during pandemic timepoints and suggested a possible beneficial effect of social support during isolation. Although the fact that neither depression nor anxiety increased across the T1–T2 span in our sample is in accord with this possibility, we also found an increase in apathy, which was not significantly associated with COVID-19 impact. Apathy may have increased in the natural course of PD progression and apparently was not staved off by potential social support in our sample. As we had no direct measure of the progression of disease in this sample (e.g., a motor exam; Hoehn and Yahr scale rating), we relied on the participants’ subjective experience. Another possibility is that the rate of the symptom change from T1 to T2 in our sample may have been steeper for apathy than for depression or anxiety, with the former being less treatable pharmacologically; PwPD isolating at home may have been able to begin or maintain psychotropic medication, resulting in stability with respect to depression and anxiety. A third, related possibility is that depression and anxiety may have been treated between T1 and T2 with selective serotonin reuptake inhibitors, which have been found to be associated with worsened apathy [6].

Gender may also have some explanatory value, as Ou and colleagues reported that apathy was associated with being male and cited other studies with similar findings. We found that apathy increased from T1 to T2 in both men and women, and the mean apathy score was consistently higher (more severe) in men, although the gender difference was not significant. As shown in Table 2, the PwPD who had participated in the BOSS-PD but not in the PD-COVID were somewhat older, more male (50% vs. 44%), had a longer duration of PD, worse non-motor and motor symptoms per the Movement Disorders Society UPDRS, and had worse apathy, depression, and anxiety. This pattern suggests that those who volunteered to participate in the follow-up study at T2 were already at an advantage in these respects during the original study at T1. We found a trend toward a significant correlation between pandemic impact (CIS) and apathy for men only (although the CIS total scores were higher in women); it is possible that, had fewer men dropped out of the original sample, the T2 data on apathy may have tracked significantly with pandemic impact.

The strength of the correlation of each of the psychiatric symptoms with pandemic impact, as indexed by the CIS, was variable, with the correlation being significant for depression and anxiety but not for apathy. For depression, the core symptoms tend to be worthlessness, thoughts of failure, guilt, and disappointment. Depression involves emotional distress and sadness. Instead of a sad mood, apathy involves a “blunted” mood [24]. In their paper [25], Brown and Pluck discussed how apathy is free from affective evaluation and may not arise from negative events or self-appraisal. Individuals with apathy lack responsiveness to both positive and negative events. This formulation may explain why anxiety and depression were correlated with subjective COVID-19 impact, but apathy was not, as the latter is associated with the lack of normal appraisal systems.

Interpretations regarding these findings should consider the limitations as well as the strengths of the study. First, the CIS is meant for a general population, and it is possible that PwPD were impacted differently by COVID-19, owing to their enhanced vulnerability related to age and chronic illness. Second, the participants who completed the surveys were primarily White, non-Hispanic individuals with high education levels. COVID-19 presumably did not impact these individuals with PD, similarly to individuals in the general population, in the same way that it would have affected those with less education and fewer overall resources. Further, the overall impact of COVID-19 in this sample was mild to moderate, and the results may have been different for individuals who were more strongly impacted by the pandemic. Third, the participants responded to an online survey, and the results may not be generalizable to in-person samples, in that there may have been a recruitment bias toward persons who were comfortable with computers and technology and had reliable internet access. It should be noted, however, that telehealth and online research methods were widely used during the pandemic, and continue to be widely used, and this method of data collection may permit many PwPD to participate in studies, particularly those whose mobility issues preclude in-person visits. The online surveys allowed us to collect data from many PwPD, and, as a further strength of the method, it is particularly notable that women were well represented in the sample, in contrast to in many in-person studies of PD. Accordingly, we were able to examine the data for potential gender effects.

## 5. Conclusions

Taken together, our results indicate that apathy, but not depression or anxiety, increased in PwPD over a time period spanning pre- to during the pandemic, and this increase was not due to the direct impact of COVID-19, age, education, disease duration, or subjective experience of motor and non-motor symptoms. By contrast, depression and anxiety did not increase during the same time period. The increase in apathy between the two timepoints may reflect the continued progression of the PD disease process. As part of the course of PD, apathy should be treated. Cognitive-behavioral therapy for depression in PwPD has been shown to also alleviate apathy [7,8], although it should be noted that apathy may also be a barrier to a response to psychotherapeutic treatment in some individuals [11]. Going forward, we also suggest that researchers be more inclusive of the full heterogeneity of PwPD, including individuals of diverse racial/ethnic backgrounds, and be attentive to the potential contributions of gender and mood to the results on the measures of apathy and on treatments of this and related psychiatric conditions.

Regarding the effects of COVID-19 on PwPD more generally, we also call attention to the fact that persons with neurological conditions, including PD, may have experienced an exacerbation of symptoms as a consequence of the pandemic [26], and that the quality of life of PwPD, as well as their care partners, were significantly impacted [27,28]. It will be of great importance to follow PwPD who experienced COVID-19 carefully post-pandemic, and particularly those PwPD who continue to experience long-lasting symptoms associated with the virus [29], including the worsening of motor symptoms, fatigue, cognitive impairment, and sleep disturbance [30]. Long-COVID may have long-term implications for non-motor as well as motor symptoms of PD and for the quality of life of PwPD and those who care for them.

## Figures and Tables

**Table 1 brainsci-13-00199-t001:** Demographic and Clinical Characteristics and Performance on Measures for Participants in Current Sample (*n* = 111) at T1 (2017–2018) and T2 (2021): Entire sample, men, and women.

	Entire Sample, T1 *n* = 111	Entire Sample, T2 *n* = 111	Men, T1 *n* = 50	Men, T2 *n* = 50	Women, T1 *n* = 61	Women, T2 *n* = 61
	Mean (SD)	Mean (SD)	Mean (SD)	Mean (SD)	Mean (SD)	Mean (SD)
Age (years)	63.8 (7.9)	67.2 (7.7)	64.4 (8.3)	67.6 (8.3)	63.3 (7.5)	66.9 (7.2)
Education (years)	16.8 (2.4)	16.8 (2.4)	17.0 (2.4)	17.0 (2.4)	16.7 (2.6)	16.7 (2.6)
Duration (years)	5.0 (4.8)	8.5 (4.8)	4.1 (3.5)	7.6 (3.5)	5.8 (5.5)	9.3 (5.6)
UPDRS_non-motor_	7.1 (3.8)	7.5 (3.7)	7.2 (4.0)	7.4 (3.2)	7.0 (3.7)	7.6 (4.0)
UPDRS_motor_	9.0 (5.9)	11.3 (6.6)	9.6 (5.8)	12.1 (5.8)	8.4 (6.1)	10.7 (7.1)
Apathy Scale	8.8 (6.4)	11.2 (6.1)	9.6 (5.8)	12.2 (6.7)	8.2 (6.9)	10.3 (5.5)
BDI-II	7.4 (6.5)	8.0 (7.0)	6.8 (4.6)	8.3 (7.2)	7.8 (7.8)	7.7 (6.8)
PAS	5.6 (6.0)	5.5 (6.4)	4.9 (5.0)	5.1 (6.5)	6.1 (6.6)	5.9 (6.4)
CIS Total	-	16.4 (3.5)	-	15.7 (3.7)	-	17.0 (3.3)

SD = standard deviation; UPDRS = Movement Disorders Society Unified Parkinson’s Disease Rating Scale; BDI-II = Beck Depression Inventory-II; PAS = Parkinson’s Anxiety Scale; CIS = Coronavirus Impact Scale.

**Table 2 brainsci-13-00199-t002:** Demographic and Clinical Characteristics and Performance on Measures for Participants in BOSS-PD and PD-COVID, and Comparison between Matched BOSS-PD + PD-COVID Participants at T1 (2017–2018) and T2 (2021).

	All BOSS-PD *n* = 347	Only BOSS-PD (Did Not Participate in PD-COVID) *n* = 217	Matched BOSS-PD + PD-COVID at T1 *n* = 123 ^a^	Matched BOSS-PD + PD-COVID at T2 *n* = 123 ^a^	Paired Samples *t*-Test between Matched Participants at T1 and T2
	Mean (SD)	Mean (SD)	Mean (SD)	Mean (SD)	* t * -value (df)
Age (years)	64.8 (8.5)	65.6 (8.7) −*n* = 214	63.8 (7.9)	67.2 (7.9)	33.6 (122) ***
Min.	40	45	40	44	
Max.	91	91	81	84	
Education(years)	16.7 (2.6)	16.7 (2.7) −*n* = 214	16.8 (2.4)	16.8 (2.4)	(no change)
Min.	10	10	10	10	
Max	21	21	21	21	
Gender (N)					
Male	165	108	54		
Female	180	107	69	54	
No answer	2	2	0	69	
Duration (years)	5.4 (4.6)	5.6 (4.5)	5.0 (4.8)	8.1 (4.8)	56.0 (122) ***
UPDRS_non-motor_	8.0 (4.3)	8.3 (4.5)	7.2 (3.8)	7.6 (3.8)	1.4 (122)
UPDRS_motor_	10.6 (7.5)	11.3 (8.0)	9.2 (6.0)	11.5 (6.8)	5.3 (122) ***
Apathy Scale ^b^	9.5 (6.5) −*n* = 293	9.8 (6.6) −*n* = 175	8.8 (6.4) −*n* = 111	11.2 (6.1) −*n* = 111	4.4 (110) ***
BDI-II	8.6 (6.8)	9.0 (6.8)	7.5 (6.5)	8.0 (6.8)	0.9 (122)
PAS	6.6 (6.8)	7.2 (7.2)	5.6 (5.8)	5.5 (6.3)	0.1 (122)
CIS Total	-	-	-	16.4 (3.4)	(T2 only)

^a^ Number of participants in PD-COVID (130) minus 7 who did not complete all measures of interest, including CIS. This number applies to all variables except where indicated otherwise. ^b^ Number who completed Apathy Scale was smaller than for other measures at both timepoints. BOSS-PD = Boston University Online Survey Study of Parkinson’s Disease; df = degrees of freedom; UPDRS = Movement Disorders Society Unified Parkinson’s Disease Rating Scale (see text); BDI-II = Beck Depression Inventory-II; PAS = Parkinson’s Anxiety Scale; CIS = Coronavirus Impact Scale. *** *p* < 0.001.

**Table 3 brainsci-13-00199-t003:** Correlations among apathy, depression, and anxiety at T1 and T2 for entire sample, women, and men.

	Apathy		Depression		Anxiety	
	T1	T2	T1	T2	T1	T2
Entire Sample						
Apathy	-	-	0.48 ***	0.56 ***	0.43 ***	0.43 ***
Depression	0.48 ***	0.56 ***	-	-	0.64 ***	0.71***
Anxiety	0.43 ***	0.43 ***	0.64 ***	0.71 ***	-	-
Women						
Apathy	-	-	0.56 ***	0.44 ***	0.47 ***	0.34 **
Depression	0.56 ***	0.44 ***	-	-	0.66 ***	0.73 ***
Anxiety	0.47 ***	0.34 **	0.66 ***	0.73 ***	-	-
Men						
Apathy	-	-	0.35 *	0.68 ***	0.42 **	0.56 ***
Depression	0.35 **	0.68 ***	-	-	0.60 ***	0.70 ***
Anxiety	0.42 **	0.56 ***	0.60 ***	0.70 ***	-	-

Notes: * *p* < 0.05; ** *p* < 0.01; *** *p* < 0.001.

## Data Availability

Data supporting the reported results can be obtained from the authors upon reasonable request, pending publication of all manuscripts from the study.

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
