# Peer review of "Changes in Apathy, Depression, and Anxiety in Parkinson’s Disease from before to during the COVID-19 Era"

_brainsci, 2023, doi:10.3390/brainsci13020199_

Round 1
Reviewer 1 Report
The authors reported an interesting study about changes several symptoms in Parkinson's Disease in the COVID-19 Era. I have some comments to the authors:
(i) In the introduction, the authors should address the lack of biomarker of disease progression in PD. This can strengthen the rationale of the study. A recent review of olfactory dysfunction as biomarker in PD should be included:
Ercoli T et al. Does Olfactory Dysfunction Correlate with Disease Progression in Parkinson’s Disease? A Systematic Review of the Current Literature. Brain Sciences 2022
(ii) Please specify inclusion and exclusion criteria for this study
(iii) The authors should include LEDD and HY values and their correlation with the other variables.
(iv) Please specify whether some of patients were on antipsychiatry drugs.
Author Response
Please see attachment, thank you.

Reviewer 2 Report
The authors conducted a study to assess the changes in apathy, depression, and anxiety symptoms for Parkinson's patients during the pandemic. Unfortunately, I don't think this article is worth. The following points will summarise my major comments on this paper, which I hope would help further improve the study.
Since the data collection time interval is 4 years, which is a relatively large interval compared to other studies that also considered the changes during COVID-19. It may need to be considered whether the changes in symptoms are because of the natural course of disease development. For example, the authors mentioned that "the increase in apathy over time in PwPD, specifically across a period of about four years, is consistent with that reported by Ou and colleagues with pre-pandemic period". What needs to be paid special attention to is that this proves that the increase in apathy is part of the development of symptoms in the natural course of Parkinson. This means that the changes in symptoms may not be impacted by COVID-19, which indicates the research topic the authors choose may be inappropriate. Thus, there have two suggestions. Firstly, if the authors have eight-year data, including 3 data collection time points (pre-pandemic twice, and during-pandemic once), the authors could compare the changes during same time interval of the pre-pandemic and pandemic period. If during 4 years time interval of the pandemic, it had bigger changes compared to 4 years time interval of pre-pandemic, the authors then can conclude that some symptoms of PwPD may correlate with COVID-19 impact. Secondly, if the authors do not have such data, it is recommended to make some slight changes to the research topic. For example, the authors could just use COVID-19 as a research background context and analyze which factors would contribute more to the changes of PwPD symptoms. The authors could use some variables that have been mentioned like "changes in routines, access to health care, changes in income, access to support, and experiences of stress".
Introduction: It is recommended to add more information about the summary of literature, gaps in literature, and explains how the research responds to some of these gaps. Especially, the authors may need to summarize the literature about the natural course of PwPD, potential influencing factors during COVID-19, and analysis of potential COVID-19 interactions with the disease's natural course.
Results: The authors mentioned "received 130 responses from March to June 2021. Data from 19 individuals who did not complete all the relevant questionnaires were excluded, for a final sample of 111 PwPD (timepoint 2, T2)", but in the table, the authors write "n=217" and "n=123" which may be wrong. Also, some text like "Only BOSS-PD " (which seems should be PD-COVID?) and some other data in the table may be wrong and do not correspond to what is reported in the article. In addition, the authors reported the r value and "Correlations among Apathy, Depression, and Anxiety", which are not shown in the tables. It is better to have the corresponding table with the text. It may be rude, but I still want to ask, do this study report a real result? I think such low-level mistakes are a sign of lack of respect for readers, reviewers, and editors and poor quality of the article. It is recommended to check and do proofreading before submitting the manuscript.
Discussion: It would be better if the authors could conclude with consideration of the wider policy or research implications of the results.
Conclusions: Please re-read the text "Taken together, our results indicate that apathy, but not depression or anxiety, increased in PwPD over a time period spanning pre- to during the pandemic, and this increase was not due to the direct impact of COVID-19, age, education, disease duration, or subjective experience of motor and non-motor symptoms", do the authors wrongly type a "not"?
Author Response
Please see attachment, thank you.

Reviewer 3 Report
The current manuscript compares the levels of depression, anxiety, and apathy before and after COVID-19 pandemic. A set of studies has studied and compared the effect of COVID on a motor and non-motor symptoms such as;
1- Schirinzi T, Di Lazzaro G, Salimei C, Cerroni R, Liguori C, Scalise S, Alwardat M, Mercuri NB, Pierantozzi M, Stefani A, Pisani A. Physical Activity Changes and Correlate Effects in Patients with Parkinson's Disease during COVID-19 Lockdown. Mov Disord Clin Pract. 2020 Aug 17;7(7):797-802.
2- Muhammad Tufail, Changxin Wu, Psychological impact of COVID-19 pandemic on Parkinson's disease patients, Heliyon,Volume 8, Issue 6, 2022.
The current manuscript found a significant increase in apathy, but not in depression or anxiety. Anxiety and depression, but not apathy, correlated with COVID-19 impact. Overall, the current manuscript needs to be reconsidreded in following points;
A- in November/2022, what is the current impact on global health and Parkinson’s disease populations.
B- The introduction needs to be shorter and more concise
C- What will this study provides for clinic or research?
D- The study compares between data at 4 years follow-up. However, there is no reporting of missing data details.
E- The discussion did not explain the potential causes of results. For example, why this significant increase in apathy, but not in depression or anxiety? and why Anxiety and depression, but not apathy, correlated with COVID-19 impact?
F- There is no reporting of statistical analysis procedure in methods
Author Response
Please see attachment, thank you. The current version incorporated changes based on these comments.

Round 2
Reviewer 3 Report
I do recommend always referring to page and line numbers of the modifications
Author Response
Thank you for your comments.
We referred to line numbers for all of our changes. I'm not sure what more the reviewer wants.